# Chemical Composition and Antimicrobial Properties of Honey Bee Venom

**DOI:** 10.3390/molecules28104135

**Published:** 2023-05-17

**Authors:** Valery Isidorov, Adam Zalewski, Grzegorz Zambrowski, Izabela Swiecicka

**Affiliations:** 1Institute of Forest Sciences, Bialystok Technical University, 15-351 Bialystok, Poland; 2Department of Experimental Physiology and Pathophysiology, Medical University of Bialystok, 15-222 Bialystok, Poland; 3Department of Microbiology, Faculty of Biology, University of Bialystok, 15-328 Bialystok, Poland; 4Laboratory of Applied Microbiology, Faculty of Biology, University of Bialystok, 15-328 Bialystok, Poland

**Keywords:** honey bee venom, stinging apparatus of bees, chemical composition of extracts, antimicrobial activity

## Abstract

Due to its great medical and pharmaceutical importance, honey bee venom is considered to be well characterized both chemically and in terms of biomedical activity. However, this study shows that our knowledge of the composition and antimicrobial properties of *Apis mellifera* venom is incomplete. In this work, the composition of volatile and extractive components of dry and fresh bee venom (BV) was determined by GC-MS, as well as antimicrobial activity against seven types of pathogenic microorganisms. One-hundred and forty-nine organic C_1_–C_19_ compounds of different classes were found in the volatile secretions of the studied BV samples. One-hundred and fifty-two organic C_2_–C_36_ compounds were registered in ether extracts, and 201 compounds were identified in methanol extracts. More than half of these compounds are new to BV. In microbiological tests involving four species of pathogenic Gram-positive and two species of Gram-negative bacteria, as well as one species of pathogenic fungi, the values of the minimum inhibitory concentration (MIC) and minimum bactericidal/fungicidal concentration (MBC/MFC) were determined for samples of dry BV, as well as ether and methanol extracts from it. Gram-positive bacteria show the greatest sensitivity to the action of all tested drugs. The minimum MIC values for Gram-positive bacteria in the range of 0.12–7.63 ng mL^−1^ were recorded for whole BV, while for the methanol extract they were 0.49–125 ng mL^−1^. The ether extracts had a weaker effect on the tested bacteria (MIC values 31.25–500 ng mL^−1^). Interestingly, *Escherichia coli* was more sensitive (MIC 7.63–500 ng mL^−1^) to the action of bee venom compared to *Pseudomonas aeruginosa* (MIC ≥ 500 ng mL^−1^). The results of the tests carried out indicate that the antimicrobial effect of BV is associated with the presence of not only peptides, such as melittin, but also low molecular weight metabolites.

## 1. Introduction

Honey bee venom (BV), also called apitoxin, is produced by the venom glands of worker bees and serves to protect the nest. This natural product contains many biologically active components belonging to different classes and groups of chemical compounds. These include proteins and free amino acids, phospholipids, biogenic amines, sugars and their derivatives, and aliphatic and aromatic hydrocarbons and their derivatives [1,2,3,4,5]. In addition, the poisonous glands of honey bees produce low molecular weight volatile organic compounds (VOCs) that act as alarm pheromones [6,7].

Bee venom has a long history of use both in traditional health practices and in official medicine. Literary sources provide information about its use in ancient Egypt over 4000 years ago, in the Hippocratic era of ancient Greece (4th century BC), and in the early European Middle Ages [8]. In the east, in China and Korea, the history of bee venom treatment also goes back many centuries [9]. One of the earliest areas of application of apitoxin is the treatment of various kinds of rheumatoid phenomena [10,11,12]. Recent studies have shown that its anti-arthritic effects are associated with anti-inflammatory action [11]. It is believed that the same property prevents the development of neurodegenerative diseases such as Alzheimer’s disease [13,14,15].

Currently, a large number of medicinal products are being produced that contain BV. This is due to the well-documented diverse biological activity of this most expensive bee product. In particular, evidence has been obtained of its antimicrobial action against both bacteria (including antibiotic-resistant strains) and pathogenic fungi [16,17,18,19,20,21,22,23]. The therapeutic effect of the external use of apitoxin in the treatment of wounds [24,25] as well as skin diseases [26] is largely associated with antimicrobial activity.

Both BV as a whole and its individual components demonstrate anti-cancer activity [10,13,23,27,28,29,30,31,32,33]. Cancer diseases are the main cause of death in the population, and many of their forms are characterized by high malignancy, metastasis, and resistance to chemotherapy [34]. Therefore, the search for alternative ways to combat these diseases with the help of natural remedies, including BV, is one of the priority areas of pharmacology [30].

The wide range of biological activity of BV and the wide range of disease manifestations that can be eliminated or mitigated by drugs based on it contrast with the degree of knowledge of its chemical composition. A consequence of insufficient chemical information about this natural product is the lack of norms and standards necessary to control its quality. Separate groups of chemical compounds of BV are currently being studied to varying degrees. Considerable attention has been paid to proteomics, and the composition of bee venom peptides has been studied to the greatest extent. However, the composition of low molecular weight metabolites has been studied to a much lesser extent, being associated with a small number of BV metabolomic studies based on the use of chromato-mass spectrometric technology [5,35]. The same applies to information about the volatile components of apitoxin; in addition to long-standing studies [6,7], we have been able to find only one report [36] on the use of modern analytical techniques (headspace solid-phase microextraction followed by gas chromatography coupled with mass spectrometry; HS-SPME/GC-MS) for the determination of VOCs in bee products, including bee venom.

This work belongs to the category of non-targeted metabolomic studies. Its purpose was to determine both the volatile components of bee venom and low molecular weight metabolites separated into two fractions by extraction with solvents of different polarity. The second goal was to characterize these fractions in terms of their antimicrobial activity.

## 2. Results and Discussion

### 2.1. Chemical Composition of Volatile Compounds

The complex social behavior of honey bees includes the collective protection of the nest, which is carried out thanks to an effective system of chemical communication with the participation of VOCs. VOCs produced by Koschevnikov’s gland are secreted by worker bees during alarm behavior [6,37]. The study of these VOCs has a long history, and it was initially found that the main component that induces aggressive behaviour in bees is 3-methylbutyl acetate (isoamyl acetate) [38]. However, this volatile compound alone elicits a weaker reaction than bee stinging apparatus extracts [39]. Further studies of the chemical composition of these extracts led to the discovery of new compounds in them [3,40,41]. In particular, Camargos et al. [41], who subjected hexane extracts to chromatography, identified 22 organic C_6_–C_36_ compounds. However, this analysis technique does not allow one to determine the most volatile components with fewer than six carbon atoms per molecule. On the other hand, extracts contain a number of low-volatility compounds with more than 20 carbon atoms.

In this study, a more modern analysis technique was used: SPME of VOCs from the vapor phase, which is in equilibrium with the object under study, and subsequent determination of the extracted compounds by GC-MS. Figure 1 shows typical VOC chromatogram profiles of dry BV and freshly extracted BV, recorded under the same conditions. As can be seen, the volatile secretions of fresh bee venom are characterized by an increased relative content of components with a retention time of more than 30 min. In the chromatograms of dry bee venom, 89 peaks of C_1_–C_19_ organic compounds of various classes were registered, the share of which in the TIC of the chromatogram was at least 0.01% of TIC, while the chromatograms of fresh venom were richer and contained 139 peaks. In the first case, 83 compounds (93%) were positively identified, and in the second case, the level of identification was lower and amounted to 77% (107 compounds). 

All registered components were divided into ten groups depending on their chemical structure. Table 1 presents the group composition of the volatile components of two samples of dry (Dv-1 and Dv-2) and two samples of freshly extracted bee venom (Fv-1 and Fv-2). Along with individual groups of compounds, their main representatives are given. A complete list of compounds registered on chromatograms is given in Appendix A.

Differences in the composition of samples of dry BV obtained in two consecutive years, one after the other, are predominantly quantitative. As can be seen from Table 1, especially strong quantitative discrepancies were found in the case of aliphatic alcohols and acids. The reasons for the observed differences are not clear. Quantitative differences in the composition of fresh BV collected from overwintered and summer bees also occur, but they are not so significant. When comparing the composition of dry and fresh BV, attention is drawn to a much larger number of individual compounds and the share in the TIC of terpenoids in fresh venom. The same applies to lactones, the content of which in the secretions of dry bee venom was insignificant, and in the case of fresh it was at the level of 5–6% TIC.

In summary, this study significantly expands the range of VOCs contained in BV, since in an earlier study using the HS-SPME/GC-MS technique, the number of components identified was 45 [36]. Some of the compounds reported in the last cited work were not found by us. These compounds were butyl nitrile, anethole, (*E*)-cinnamaldehyde, 1-methyl-3-cyclohexene-1-carbaldehyde, cyclooctanol, geranyl acetone, geranic acid, and tetradecanoic acid.

### 2.2. Chemical Composition of Extractive Compounds

The investigated BV preparations were subjected to successive extraction with low-polarity diethyl ether and highly polar methanol. This approach makes it possible to simplify chromatograms and reduce the likelihood of mutual overlapping of chromatographic zones of different compounds, as well as masking of minor peaks by peaks of high intensity. In addition, this made it possible to compare the antimicrobial activity of the fractions, which include compounds of different polarity.

#### 2.2.1. Chemical Composition of Extracts with Diethyl Ether

In the chromatograms of extracts of dry BV, 90 peaks of C_2_–C_42_ organic compounds were registered, while in extracts of fresh venom, 152 compounds were registered with the same range of molecular weights. Table 2 shows the relative content of the nine groups along with the main representatives of each of them. A complete list of substances is given in Appendix A.

The most numerous group is formed by linear C_19_–C_33_ alkanes and C_21_–C_35_ alkenes, which account for 32–45% TIC and 25–27% TIC in extracts of dry and fresh bee venom, respectively. The esters of oleic and palmitic acids and aliphatic C_18_–C_24_ alcohols form the second largest contribution to the TIC of the chromatogram of the dry BV extract.

Aliphatic acids form a large group in extracts of fresh venom, with oleic acid predominating quantitatively. The content of acids in dry BV turned out to be much lower, and the main one was citric acid. A significant share in all four extracts also falls on the share of aliphatic alcohols. In general, ether extracts are characterized by a high content of lipid components (alkanes and alkenes, aliphatic long-chain alcohols and acids, and glycerides) and a low content of aromatic compounds.

#### 2.2.2. Chemical Composition of Methanol Extracts

In the chromatograms of methanol extracts of dry BV, 137 peaks with a relative content of at least 0.01% TIC were recorded, while in the case of fresh venom there were 161 peaks. In total, 201 compounds were registered in all four samples. The group composition of the extracts is given in Table 3, and the full composition is given in Appendix A.

These extracts were characterized by the largest number of unidentified compounds (most of them were minor components). In all four extracts, the largest number of identified compounds belong to the group of carbohydrates and related compounds (sugar alcohols and acids). The predominant carbohydrates in them were monosaccharides and disaccharide trehalose, while sucrose and trisaccharides (1-kestose, erlose, and melizitose) were found only in dry BV extracts.

The second largest group is formed by aliphatic acids, predominantly polar hydroxy acids, the main one of which was citric acid, but only in dry BV extract. Interestingly, in these methanol BV extracts, citric acid was the main individual compound, accounting for 29.5% and 27.6% of TIC. It was previously found by the authors in dried BV obtained by electrical stimulation [5]. The number of amino acids identified in the analyzed samples differed: in fresh venom samples there were more than 20 amino acids, and their composition almost overlapped with that published by the authors [5]. However, we identified 14 amino acids in dry BV. Their list includes both proteinogenic and non-proteinogenic amino acids such as sarcosine, β-alanine, hydroxyproline, and GABA. Quantitatively, the predominant components were proline and alanine, which is consistent with the data of [5].

All analyzed samples also contained a large number of other nitrogen-containing compounds, the main one of which was histamine. The extracts contained nitrogen compounds related to catecholamines (neurotransmitters dopamine, norepinephrine, and serotonin), purines (adenine, xanthine, hypoxanthine, and uric acid), nucleosides (guanosine, uridine, pseudouridine, inosine, and cytidine), and pyrimidines (5-methylcytosine, and uracil). All four extracts contained diamines, putrescine, and cadaverine, as well as *N*-acetylputrescine. Although the latter compound was found by the authors [5] in dried BV, the content of free putrescine and cadaverine in bee venom, to the best of our knowledge, has not been previously reported.

### 2.3. Antimicrobial Activity of BV and Extracts

Much attention has been paid to the study of the antimicrobial action of bee venom [16,17,18,19,20,21,22]. These properties are due to the presence of peptide compounds, phospholipase А2, MCD (mast cell degranulating) peptide, and melittin. The antibiotic effect of BV is related to the ability of melittin and MCD peptide to damage cell membranes, including the protective membranes of Gram-negative bacteria such as *E. coli* and various *Pseudomonas* species [21,42,43,44,45]. One of the goals of this study was to elucidate the role of low molecular weight metabolites in the formation of the antimicrobial activity of BV. In experiments aimed at solving the problem, the values of the MIC and MBC/MFC for BV samples and extracts obtained from them were determined. The values given in Table 4 indicate a very high activity of both tested samples of BV against Gram-positive bacteria and a pathogenic fungus. Interestingly, in the case of Gram-negative bacteria, *E. coli* appeared to be more sensitive to the action of BV.

The least active was the ether extract, for which the MIC values were almost two orders of magnitude greater than for the whole BV and significantly higher than for the methanol extract. However, the inhibitory ability of the ether extract against Gram-positive bacteria and the fungus *C. albicans* does not seem insignificant when compared with a similar characteristic for such a recognized natural antibiotic as propolis. For example, different types of propolis show MIC values for the dangerous honey bee pathogen *P. larvae*, in the range of 8–125 µg mL [46,47], i.e., orders of magnitude higher than the ether extracts from BV. The same difference in MIC values is observed between the action of ether extracts and different types of propolis on other microbes shown in Table 4 [47,48].

The results obtained indicate that the antimicrobial activity of BV is associated not only with the action of the peptides contained in it, but at least in part with low molecular weight metabolites produced by venom glands.

## 3. Materials and Methods

### 3.1. Chemicals and Materials for Microbiological Research

Extraction of the test material was carried out with diethyl ether and methanol (POCH, Gliwice, Poland). Pyridine, bis(trimethylsilyl)trifluoroacetamide (BSTFA) with the addition of 1% trimethylchlorosilane (TMC) used for derivatization, a calibration mixture of C_8_–C_40_ *n*-alkanes, and an SPME fiber holder and SPME fiber assembly divinylbenzene/Carboxen/polydimethylsiloxane (PDMS) were obtained from Sigma-Aldrich (Poznan, Poland). All of the microbiological media used in the study were supplied by Oxoid Ltd. (Basingstoke, UK). A 0.22 μm pore-size Rotilabo^®^ syringe filter was supplied by Carl Roth GmbH and Co (Karlsruhe, Germany).

### 3.2. Bee Venom Preparations

Two samples of dry BV (Dv-1 and Dv-2) were kindly provided in March 2021 and February 2022 by the beekeeping company Sądecki Bartnik^®^ (Stróże, Poland). According to the supplier, these samples were obtained by electrical stimulation of honey bees (race of bees not reported) in the summers of 2020 and 2021, respectively.

Fresh samples of bee venom (Fv-1 and Fv-2) were obtained by removing the stinger from Apis mellifera carnica anaesthetized at −20 °C and separating the venom reservoir from it, as described in the manual [49]. Sampling was carried out twice, from overwintered bees at the end of March 2022 and in August of the same year. Each time, the stinging apparatus was removed from approximately 200 bees by placing it in a 16 mL glass headspace vial.

### 3.3. Determination of Volatile Compounds

The determination of the composition of volatile components (VOCs) of dried and freshly extracted bee venom was carried out according to the previously described procedure, using HS-SPME/GC-MS [50,51,52]. According to the results of the experiments described in the cited papers to determine the composition of VOCs of various natural materials from the available range of sorption fibers, the choice was made to use divinylbenzene–Carboxen–PDMS fiber (DVB/CAR/PDMS; Supelco/Sigma-Aldrich, Poznan, Poland).

One gram of dry bee venom was weighed into 16 mL vials for headspace analysis, closed with a lid, and placed in a thermostat heated to 40 °C. A sorption fiber was introduced into the vial, piercing the silicone membrane in the cap with a protective needle. The duration of exposure of the fiber in the gas phase of the flask was 1 h. After that, the fiber was placed for 10 min in the injector of the GC-MS apparatus heated to 180 °C. In the case of VOC analysis of fresh bee venom collected as described above, the sorption fiber was exposed at room temperature (22 ± 1 °C) for 0.5 h.

The separation of VOCs adsorbed on the fiber was carried out on an HP7890A gas chromatograph equipped with a 5975C VL MSD triple-axis detector (Agilent Technology, Santa Clara, CA, USA). The GC was fitted with an HP-5ms capillary column (30 m × 0.25 mm i.d., 0.25 μm film thickness). The carrier gas (He) flow rate through the column was 1 mL min^−1^. The initial temperature was 40 °C and rose to 220 °C at a rate of 3 °C min^−1^. A split/splitless injector was operated at 200 °C in splitless mode. The mass spectrometric detector acquisition parameters were as follows: the transfer line, MS source, and MS quadrupole temperatures were 280, 230, and 150 °C, respectively. Electron impact mass spectra were obtained at an ionization energy of 70 eV. Detection was performed in full scan mode from 39 to 300 a.m.u.

The contribution of each peak to the total ion current (TIC) of the chromatogram was calculated from the integration results. The retention indices (*RIs*) of the separated components were calculated by taking into account their retention times, as well as the retention times of normal C_6_–C_18_ alkanes recorded under the above conditions.

### 3.4. Determination of Extractive Compounds

Dry bee venom, as well as fresh bee venom extracted from bees together with the stinging apparatus, was successively extracted three times with diethyl ether and methanol. The combined ether and methanol extracts passed through a paper filter were evaporated to dryness in glass cups in a fume hood. Approximately 5 mg of the precipitate remaining on the walls was transferred into a 2 mL vial and dissolved in 220 µL of dry pyridine; 80 µL of BSTFA was added, and the mixture was heated to 60 °C to complete the derivatization process.

Separation of the silanized components was carried out on the above-mentioned capillary column at a carrier gas velocity of 1 mL min^−1^. Sampling of 1 μL of the reaction mixture was carried out using an Agilent 7693A autosampler. The injector was heated to a temperature of 300 °C and worked in a split (1:20) mode. The initial temperature of the column thermostat was 50 °C and increased to 320 °C at a rate of 3 °C min^−1^. The temperatures of the ion source and quadrupole were 230 °C and 150 °C, respectively. Mass spectra were obtained at an ionization energy of 70 eV. Detection was carried out in the mass range from 41 to 650 a.m.u.

The *RIs* of the mixture components were calculated using their retention times, as well as the retention times of С_10_–С_40_ *n*-alkanes recorded under the above conditions.

### 3.5. Component Identification

Separated components were identified by their mass spectra using an automatic GC/MS processing system equipped with an National Institute of Standards and Technology NIST 14 electron ionization mass spectra library. *RIs* contained in collections [53,54,55] were used as an independent analytical parameter. Mass spectrometric identification was considered reliable if its results were confirmed by experimental *RI* values, that is, if their deviation from those published in databases did not exceed ±10 u.i. If the results of mass spectrometric identification were not confirmed by *RI* values due to their absence in the available databases or if the discrepancy exceeded 10 u.i., the identification was considered tentative.

### 3.6. Component Quantification

The precision of the method was expressed by the relative standard deviation (RSD). Peak areas determined by HS-SPME triplicate analysis of volatile components and analysis of ether and methanol extracts of a dry sample of bee venom (DV-1) were used to calculate RSD values that did not exceed 7% for components with a relative content of more than 5% of TIC but reached 30% for components whose content did not reach 1%.

### 3.7. Determination of Antimicrobial Activity of Bee Venom and Extracts

Dry BV, as well as diethyl ether and methanol extracts obtained from it, was tested against microorganisms originating from the *American Type Culture Collection* (LGC Standards Sp. z o.o., Lomianki, Poland): Gram-positive bacteria *Staphylococcus aureus* ATCC 6538, *Paenibacillus larvae* ATCC 9545, *Bacillus cereus* ATCC 10987, and *Bacillus subtilis* ATCC6633 and Gram-negative bacteria *Escherichia coli* ATCC 11229 and *Pseudomonas aeruginosa* ATCC 19582, as well as the fungus *Candida albicans* ATCC 90029. All microorganisms stored in a 1:1 mixture of Luria–Bertani (LB) broth and glycerol were inoculated into either nutrient agar (bacteria) or Sabouraud agar (*C. albicans*) and incubated overnight at 37 °C.

The antibacterial activity of the tested preparations was assessed by determining the minimal inhibitory concentration (MIC) in accordance with the Clinical and Laboratory Standard Institute (CLSI) protocols [56]. Bee venom extracts were dissolved in DMSO at a concentration of 8 mg mL^−1^, filtered with a 0.22 μm pore-size Rotilabo^®^ syringe filter and serially twofold diluted in Mueller–Hinton broth, ranging from 4000 to 0.0002 μg mL^−1^, in a U-shaped 96-well microtiter plate with a final volume 100 μL. The bacteria were cultured overnight in Mueller–Hinton broth at 37 °C with shaking at 200 rpm and then suspended to a final optical density of 0.2–0.3 at 600 nm wavelength measured with a V-670 spectrophotometer (Jasco, Tokyo, Japan). For the assay, 100 μL of the bacterial suspensions was added to each well in the microtiter plate containing diluted bee venom extracts and incubated overnight at 37 °C. In order to obtain comparable data, all the bacteria were treated under the same conditions. The MIC values were determined as the lowest concentration of the extracts in the wells with no bacterial growth observed visually. All the tests were carried out in quadruplicate, and the results were averaged.

In addition, the minimal bactericidal concentration (MBC) and minimal fungicidal concentration (MFC) of the extracts were assessed. For this purpose, 5 µL of the overnight culture from each well in the microtiter plate with extracts of concentration equal to and higher than the MIC value was inoculated onto BHI (Brain Heart Infusion) agar with the use of a sterile plastic spreader and incubated overnight at 37 °C. The MBC/MFC values were determined as the lowest concentration of the extracts in the wells with no bacterial growth on plates observed visually. All the tests were carried out in quadruplicate, and the results were averaged.

As a positive control, microorganisms cultured in the Mueller–Hinton broth and on the BHI agar without the bee venom extracts were applied. Mueller–Hinton broth supplemented with 10% dimethyl sulfoxide (DMSO) was used as solvent control, while Mueller–Hinton broth with 10% DMSO and extracts was used as the bee venom extract control. The MIC and MFC values for *C. albicans* were assessed as above, but with the application of Sabouraud broth and Sabouraud agar instead of Mueller–Hinton broth and BHI agar, respectively.

## 4. Conclusions

This study significantly expands the range of volatile organic compounds found in bee venom by using a more modern method of analysis and for the first time reports the composition of low molecular weight compounds extracted from it with solvents of different polarity. Further research, taking into account the likely features associated with the geographical and seasonal origin of BV, can serve to further expand these lists.

Microbiological testing has confirmed the antimicrobial activity of extracts that do not contain the peptides to which it is usually attributed. It seems interesting that it manifests itself at a very low level of concentration. The search for natural antimicrobial compounds is very relevant now, in the context of the observed and aggravated “antibiotic crisis” caused by the emergence of antibiotic-resistant strains of many dangerous pathogens [57,58]. Antibiotic resistance leads to higher medical costs, prolonged hospital stays, and increased mortality. In light of this, a deeper study of the antimicrobial properties of BV with a view to their use in treatment is of interest. At the same time, the potential toxicity of the constituents of bee venom to normal non-target cells, which may interfere with its medical use, as is the case for many animal venoms, needs to be studied [59,60,61].

## Figures and Tables

**Figure 1 molecules-28-04135-f001:**
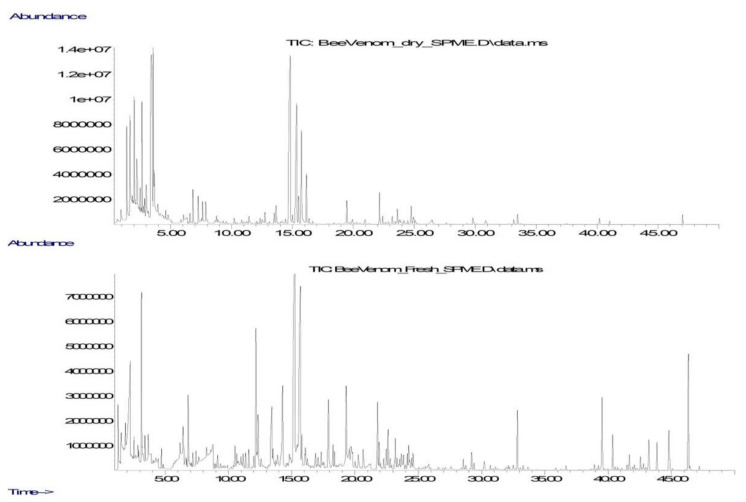
Chromatographic profile of volatiles in dry bee venom (**top**) and fresh extracted venom (**bottom**).

**Table 1 molecules-28-04135-t001:** Group composition (% of TIC) of VOCs from dry and fresh venom of honey bees.

Groups of Compounds	Dried Venom		Fresh Venom	
	Dv-1	Dv-2	Fv-1	Fv-2
Aliphatic carbonyls, including:	26.67	8.14	5.51	19.13
- acetone	0.59	0.50	0.15	2.30
- 2-butanone	trace *	0.18	1.18	1.56
- 2-pentanone	1.37	trace	0.40	0.73
- 2-heptanone	16.08	0.49	0.76	1.53
- 2-nonanone	1.99	1.57	1.19	10.94
- 2-undecanone	- **	-	0.57	0.76
- isobutanal	1.88	0.02	-	-
- isopentanal	0.89	0.71	-	-
- hexanal	1.73	0.01	-	-
- nonanal	0.71	trace	-	-
- (2*E*)-decenal	-	-	0.30	0.23
Aliphatic alcohols, including:	1.85	50.37	15.66	20.11
- ethanol	1.46	trace	0.33	0.26
- isopentanol	0.40	28.90	3.56	5.10
- (3*Z*)-hexen-1-ol	-	0.73	0.34	0.23
- 2-heptanol	trace	0.21	0.15	0.40
- 2-nonanol	trace	0.57	5.79	8.46
- 2-ethyl-1-hexanol	-	-	1.84	1.38
- 2-undecanol	-	-	0.42	0.40
- (2*E*)-decen-1-ol	-	-	0.55	0.79
Aliphatic acids, including:	32.70	4.53	12.50	9.24
- formic acid	2.37	trace	-	-
- acetic acid	6.60	3.00	7.97	7.83
- isobutyric acid	-	-	0.60	0.91
- butyric acid	trace	trace	0.59	0.50
- hexanoic acid	1.43	0.30	0.76	-
- octanoic acid	20.72	0.05	0.93	0.50
- 2-octenoic acid	0.50	trace	0.50	0.15
Aliphatic esters, including:	12.49	14.43	9.13	9.80
- ethyl acetate	0.95	trace	0.54	trace
- isoamyl acetate	trace	0.13	1.45	4.92
- isoamyl isobutanoate	trace	0.76	0.25	0.38
- ethyl 2-methylvalerate (manzanate)	-	-	1.23	1.51
- isoamyl pentanoate	-	-	0.45	0.83
- isoamyl 3-methy-2-butenoate	-	2.67	1.15	0.14
- isoamyl octanoate	trace	trace	0.08	0.08
- isoamyl benzoate	-	0.23	0.16	0.29
- isopropyl tetradecanoate	-	-	0.60	1.09
- ethyl octanoate	10.59	-	-	-
Aromatics, including:	18.34	1.46	9.45	5.51
- toluene	3.40	1.15	1.93	1.42
- *p*-cymene	trace	trace	0.64	0.28
- benzaldehyde	2.62	-	0.15	0.11
- acetophenone	0.33	trace	0.43	0.58
- 1-phenyl ethanol	-	-	0.12	0.21
- 2-phenyl ethanol	trace	trace	1.39	0.82
- cresol	-	-	1.22	1.29
- *p*-ethylguaiacol	-	-	0.37	0.37
- methyl benzoate	1.88	-	-	-
-methyl salicylate	9.52	-	-	-
Terpenoids, including:	3.91	0.56	12.60	12.48
- α-pinene	1.78	trace	2.06	2.39
- β-pinene	0.54	-	0.47	trace
- 3-carene	0.36	trace	0.28	0.32
- limonene	0.56	0.35	2.90	3.12
- dihydromyrcenol	-	-	2.37	3.08
- camphor	--	-	0.28	0.33
- borneol	-	-	0.32	0.37
- bornyl acetate	-	-	0.64	0.38
- β-caryophyllene	trace	0.21	0.06	0.17
- γ-cadinene	-	-	0.07	0.11
Alkanes and alkenes, including:	2.95	19.66	14.66	8.45
- *n*-hexane	2.62	-	-	-
- *n*-decane	-	trace	-	0.48
- *n*-dodecane	-	trace	2.00	0.26
- *n*-tridecane	0.33	0.19	0.16	0.28
- *n*-tetradecane	-	-	0.17	0.11
- *n*-pentadecane	-	0.49	1.44	1.36
- *n*-heptadecane	-	0.18	1.47	1.95
- 1-pentadecene	-	16.99	0.17	0.10
- nonadecene	-	0.45	5.51	2.87
Lactones, including:	0.30	trace	5.53	5.68
- valerolactone	trace	-	0.16	-
- γ-caprolactone	trace	-	1.71	2.56
- γ-heptalactone	-	trace	2.05	2.05
- γ-octalactone	0.30	trace	1.05	0.95
Other, including:	1.42	1.85	-	2.35
- pyridine	0.46	0.52	-	0.71
- 2,3-butanediol	-	-	-	0.24
- 2-pentylfuran	0.20	-	-	-
- dimethyl sulfide	-	0.81	-	-
- acetoin	-	0.52	-	trace
NN	0.55	1.42	6.38	7.25

* less than 0.01% TIС, ** component not found.

**Table 2 molecules-28-04135-t002:** Group composition (% of TIC) of ether extracts from dried and fresh bee venom.

Groups of Compounds	Dried Venom		Fresh Venom	
	Dv-1	Dv-2	Fv-1	Fv-2
Aliphatic Alcohols, Including:	16.90 (12) *	13.37 (7)	18.04 (16)	19.00 (11)
- isopentanol	- **	-	0.93	0.16
- oleyl alcohol	0.10	0.09	0.29	0.23
- octadecanol	0.08	0.06	0.27	0.12
- (9*Z*)-eicosen-1-ol	14.75	11.71	12.40	16.16
- 1-docosanol	0.06	-	0.33	0.57
- 1-tetracosanol	0.19	0.16	0.47	0.35
- 1-hexacosanol	0.23	0.17	0.37	0.33
Aliphatic acids, including:	3.23 (7)	4.90 (9)	27.31 (37)	31.90 (19)
- lactic	0.05	0.05	0.48	trace ***
- citric	1.88	3.79	-	-
- palmitic	0.34	0.29	2.40	4.93
- α-linoleic	-	-	2.12	1.94
- oleic	0.80	0.65	12.15	20.14
- stearic	0.06	0.06	3.18	2.76
Aliphatic esters, including:	17.31 (31)	20.77 (6)	8.07 (7)	6.17 (7)
- octadecyl oleate	2.59	0.22	0.62	0.22
- eicosenyl oleate	13.18	17.42	1.89	1.37
- eicosyl oleate	1.54	1.51	0.92	1.11
- tetracosyl palmitate	-	0.39	3.42	1.14
Glycerol & glycerides, including:	5.47 (4)	8.22 (5)	1.02 (2)	0.58 (2)
- glycerol	0.36	0.13	0.07	trace
- 1-hexadecyl glycerol	0.07	0.07	-	-
- 1-eicosyl glycerol	1.86	1.43	0.95	0.58
- 2-eicosyl glycerol	3.16	2.43	-	-
Aromatic compounds, including:	1.76 (2)	1.05 (2)	4.62 (10)	0.58 (4)
- benzoic acid	-	-	1.60	0.30
- *p*-hydroxybenzoic acid	-	-	0.28	0.22
- gallic acid	1.52	0.88	-	-
- ellagic acid	0.24	0.17	-	-
Sterols, including:	0.23 (1)	0.10 (1)	2.89 (3)	6.51 (4)
- 24-methylenecholesterol?	-	-	1.94	3.26
- β-sitosterol	-	-	trace	1.29
- avenasterol	0.23	0.10	0.94	1.65
Alkanes and alkenes, including:	45.30 (25)	32.07 (30)	25.82 (33)	27.10 (34)
- *n*-pentacosane	3.41	2.63	2.16	1.75
- *n*-heptacosane	5.51	4.27	3.55	3.39
- *n*-nonacosane	3.28	2.46	3.35	2.31
- *n*-hentriacotane	2.23	1.76	2.40	2.31
- 7-hentriacontene	4.42	3.46	0.58	2.45
- 9-hentriacontene	3.34	2.62	3.70	2.30
Other compounds, including	3.90	12.54	3.78	2.84
- ethylamine	-	8.44	0.30	0.47
- uracil	-	-	0.04	-
- indole-3-acetic acid	-	-	0.06	-
NN	5.90 (10)	6.98 (15)	8.45 (27)	5.90 (10)

* the number of components in a particular group is given in parentheses; ** component not found; *** less than 0.01% TIС.

**Table 3 molecules-28-04135-t003:** Group composition (% of TIC) of methanol extracts from dry and fresh honey bee venom.

Groups of Compouns	Dry Venom		Fresh Venom	
	Dv-1	Dv-2	Fv-1	Fv-2
Aliphatic Alcohols, Including:	3.56 (2) *	2.15 (3)	5.51 (12)	5.58 (2)
- glycerol	0.42	0.15	2.25	2.24
- (9*Z*)-eicosen-1-ol	3.14	0.97	1.82	3.34
- oleyl alcohol	- **	0.04	-	-
Aliphatic acids, including:	33.60 (18)	28.62 (18)	17.17 (26)	12.61 (17)
- lactic	0.49	0.22	4.00	0.38
- glycolic	0.11	0.05	-	0.10
- glyceric	0.62	0.04	-	-
- succinic	0.67	0.16	1.04	0.52
- malic	0.34	0.23	0.14	0.24
- citric	29.51	27.60	0.09	-
Aminoacids, including:	3.97 (10)	1.13 (14)	22.81 (21)	19.10 (24)
- glycine	0.14	0.08	1.12	0.76
- alanine	0.91	0.07	3.52	2.80
- prolinę	2.05	0.45	4.30	4.98
- 5-oxoproline	0.77	0.09	0.10	1.28
- β-alanine	0.10	0.15	1.59	0.73
- glutamine	trace	0.10	0.74	1.65
- lysine	trace	0.02	0.49	1.05
- hydroxyproline	-	-	0.09	0.92
- γ-aminobutanoic (GABA)	-	0.01	Trace ***	0.27
Other N-containing compounds, including:	12.10 (15)	23.71 (21)	7.63 (13)	7.36 (16)
- 2-aminoethanol	0.12	0.10	0.40	trace
- putrescine	0.72	0.63	1.54	1.49
- cadaverine	0.33	0.56	trace	trace
- histaminę	5.74	18.25	1.81	0.35
- uridine	0.99	0.32	-	-
- uracil	-	-	0.40	0.43
- adenine	-	-	trace	0.29
- serotonin	1.91	0.80	-	0.56
- inosine	1.91	0.80	-	0.56
P-containing compounds, including:	12.23 (2)	7.17 (2)	5.48 (5)	9.29 (5)
- H_3_PO_4_	11.54	6.89	2.48	5.13
- α-glycerophosphate	0.69	0.28	2.33	3.31
- *myo*-inositol phosphate	-	-	-	0.72
Carbohydrate & related compounds, including:	27.20 (21)	32.37 (30)	30.34 (44)	38.50 (35)
- α- and β-fructose	1.96	14.26	4.58	7.23
- α- and β-glucose	0.14	4.11	12.13	16.51
- gluconic acid	0.51	4.05	2.81	3.18
- mannitol	0.04	0.59	0.77	1.34
- glucitol	trace	0.24	2.00	2.76
- *myo*-inositol	0.85	0.44	0.31	0.21
- sucrose	15.85	2.97	-	-
- trehalose	2.91	1.52	0.26	1.02
- 1-kestose	0.25	0.07	-	-
- erlose	1.36	0.02	-	-
- melizitose	0.77	trace	-	-
Other compounds, including:	3.13 (5)	0.12 (4)	1.14 (5)	3.25 (7)
- 1-*O*-eicosyl glycerol	0.62	0.04	-	0.62
- β-sitosterol	-	-	0.16	0.14
- avenasterol	-	-	0.37	-
- unidentified P-containing compound	-	-	0.30	-
- 9-hentriacontene	0.22	0.03	0.10	-
- 7-hentriacontene	0.14	0.03	0.21	-
NN	4.24 (12)	4.75 (26)	5.59 (27)	4.31 (15)

* the number of components in this group is given in brackets; ** component not found; *** less than 0.01% TIC.

**Table 4 molecules-28-04135-t004:** Antimicrobial activity of bee venom samples and extracts obtained from them.

Sample	Gram-Positive Bacteria				Gram-Negative Bacteria		Fungus
	*P. larvae* ATCC 9545	*S. aureus* ATCC 6538	*B. cereus* ATCC 10987	*B. subtilis* ATCC 6633	*P. aeruginosa* ATCC 19582	*E. coli* ATCC 11229	*C. albicans* ATCC 90029
MIC, ng mL^−1^							
Bee venom Bv-1	1.91	7.63	7.63	0.12	>500	122.07	30.52
- ether extract	122.07	500	-	125	>500	>500	488.28
- methanol extract	31.25	122.07	122.07	7.63	500	500	125
Bee venom Bv-2	0.48	1.91	0.48	0.12	>500	7.63	7.63
- ether extract	31.25	125	125	125	500	500	7.63
- methanol extract	7.81	31.25	125	0.49	>500	125	1.95
MBC/MFC, ng mL^−1^							
Bee venom Bv-1	7.63	122.07	122.07	0.48	>500	488.28	122.07
- methanol extract	488.28	>500	488.28	30.52	>500	>500	>500
Bee venom Bv-2	1.91	1.91	1.91	0.48	>500	30.52	30.52
- methanol extract	31.25	31.25	500	30.52	>500	-	500

## Data Availability

This study did not report any data.

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
