# Peer review of "Chemical Composition and Antimicrobial Properties of Honey Bee Venom"

_molecules, 2023, doi:10.3390/molecules28104135_

Round 1

Reviewer 1 Report (Previous Reviewer 1)

It seems that all the suggestions have been taken into consideration and that manuscript has been improved.

Author Response

Probably no response required as the reviewer did not make any further comments.

Reviewer 2 Report (Previous Reviewer 3)

Authors changed and improved their paper substantially. There are only few minor remarks that could be dealt with.

There are different fonts all over the paper so please unify that.

Reference number 34 is not cited in the text?

If possible, please provide better resolution for Figure 1 (maybe it just seems blurry in my version).

Author Response

Indeed, reference 34 was not mentioned in the text of the article. This omission has been corrected. A note regarding the different fonts has also been taken into account.

Reviewer 3 Report (New Reviewer)

Dear authors,

After reading your manuscript, I realized that you examined the chemical Composition and Antimicrobial Properties of Honey Bee Venom. The composition and antimicrobial properties of Apis mellifera venom are incomplete and very limited. The issue that you dealt with is well described in the paper, and your work complements previous knowledge and opens up the possibility for further research. The paper is technically and clearly written. The discussion is clearly written. The methodological details are explained well. Also, the tables are very clearly presented.

Remarks: The linguistic quality of the paper should be revised more carefully once more by an English expert.

The title summarizes the organization of the work well. On the contrary, the abstract deserves further study, especially for the part relating to the results

The Introduction does not indicate the status of current knowledge. Moreover, there doesn’t seem to be a clear research hypothesis formulated. 

The experimental design is appropriate to resolve the stated objectives of the study. The experimental techniques are appropriate to resolve the stated objectives of the study.

If the suggested changes and information are correctly clarified, I believe that the manuscript is suitable for publication in Journal. Certainly, the results are important to scientific literature. My recommendation is to publish the paper after the MINOR revision of the manuscript.

Sincerely,

The linguistic quality of the paper should be revised more carefully once more by an English expert.

Author Response

Comment 1. The linguistic quality of the paper should be revised more carefully once more by an English expert.

Answer: The manuscript of the article underwent linguistic editing by an English-speaking specialist. We hope that possible minor language errors will be corrected when preparing the article for publication, since the rules for authors state: «MDPI provides minor English editing by native English speakers for all accepted papers, included in the APC»

Comment 2 The title summarizes the organization of the work well. On the contrary, the abstract deserves further study, especially for the part relating to the results.

Answer: We do not understand what this reviewer's remark is about: whether it means the need to introduce additional information about the results of the study, or vice versa, to reduce it. One of the comments of the following reviewer also concerns the abstract, namely the number of words in it.

Comment 3. The Introduction does not indicate the status of current knowledge. Moreover, there doesn’t seem to be a clear research hypothesis formulated.

Answer: Probably, the reviewer did not pay attention to the fact that the introduction says that the wide practical use of bee venom contrasts with the degree of knowledge of its chemical composition (lines 67-69). The same paragraph tells what gaps there are in our knowledge about this product.

In the last paragraph of the "Introduction" section, the tasks set by us in this study are formulated. It seems to us that this is enough and there is no need to formulate any research hypothesis.

Reviewer 4 Report (New Reviewer)

In this manuscript, the authors Isidorov et al. evaluated the composition and antimicrobial properties of honey bee venom.

This manuscript is interesting and very complex in terms of the methods used: determination of volatile compounds, determination of extractive compounds, component quantification, antimicrobial activity.

I believe that the article can be published, with some minor recommendations:

The abstract is too long, with too many details. In Instructions for Authors, it is recommended that the abstract should be a total of about 200 words maximum, summarize the article's main findings and indicate the main conclusions or interpretations.

At point 2.7. it is not mentioned from which supplier the strains of microorganisms were obtained.

There are some abbreviations that are not explained when they first appear in the text.

The company producing the laboratory equipment must be mentioned.

In Table 4, where are the results for MBC/MFC for ether extracts?

At the Conclusions there should not be References but only the authors' own conclusions.

Author Response

Comment 1. The abstract is too long, with too many details. In Instructions for Authors, it is recommended that the abstract should be a total of about 200 words maximum…

Answer: The abstract in the original version of the article fit into the length specified in the rules for authors (<200 words). However, one of the reviewers suggested that additional information be provided in it: The authors should make the abstract more informative by providing the results of MIC. At least, the MIC for the microorganisms more susceptible to the extract. The authors should highlight in the abstract the most active extract. This requirement has been met, resulting in an increase in the length of this section. We are in difficulty and do not understand how to combine, on the one hand, the requirements of the editorial board, and, on the other hand, the proposals of the reviewers.

Comment 2. At point 2.7 it is not mentioned from which supplier the strains of microorganisms were obtained.

Answer: Added microbiological culture supplier details to section 2.7

Comment 3. There are some abbreviations that are not explained when they first appear in the text.

Answer: As a result of a thorough revision of the text, it was possible to find some abbreviations that, in our opinion, do not need to be deciphered. However, as recommended by the reviewer, we present it for: National Institute of Standards and Technology, NIST (line 157), Luria-Bertani (LB) broth (line 180), Brain Heart Infusion, BHI (line 200), dimethyl sulfoxide, DMSO (line  207).

Comment 4. The company producing the laboratory equipment must be mentioned.

Answer: It seems to us that we have given the names of the manufacturers of the main used analytical equipment: the GC-MS apparatus and the V-670 spectrophotometer (Jasco, Japan).

Comment 5. In Table 4, where are the results for MBC/MFC for ether extracts?

Answer: In the case of essential extracts, MBC/MFC values were not determined.

Comment 6. At the Conclusions there should not be References but only the authors' own conclusions.

Answer: We apologize, but we have never encountered such restrictions anywhere. It seems to us that the authors have the right to support their conclusions with references to the work of other researchers.

This manuscript is a resubmission of an earlier submission. The following is a list of the peer review reports and author responses from that submission.

Round 1

Reviewer 1 Report

The article presents the Chemical Composition and Antimicrobial Properties of Honey Bee Venom. Honey bee venom has a great medical and pharmaceutical importance. In this study the composition of volatile and extractive components of dry and fresh bee venom was determined, as well as antimicrobial activity against seven types of pathogenic microorganisms. Before recommending this article for publication some suggestions are needed as follow:

In Figure 1. the description is “Chromatogram of volatiles in dry bee venom (top) and fresh extracted venom (bottom)” but the images for chromatograms are inversed.

In Tabel 1. E and Z should be italicized, 2-Pentylfuran should be 2-pentylfuran.

In Tabel 2. n should be italicized.

In Table 3. H3PO4 should be H3PO4.

N-acetylputrescine should be N-acetylputrescine.

I would also like to suggest a Conclusions section and an Abbreviations list/table.

Throughout the entire manuscript there are different font types used for the body text.

Author Response

Comment 1: In Figure 1. the description is “Chromatogram of volatiles in dry bee venom (top) and fresh extracted venom (bottom)” but the images for chromatograms are inversed.

In Tabel 1. E and Z should be italicized, 2-Pentylfuran should be 2-pentylfuran.

In Tabel 2. n should be italicized.

In Table 3. H3PO4 should be H3PO4.

N-acetylputrescine should be N-acetylputrescine.

Throughout the entire manuscript there are different font types used for the body text

Reply: All these comments have been taken into account and appropriate corrections have been made to the text of the article.

Comment 2: I would also like to suggest a Conclusions section and an Abbreviations list/table.

Reply: In accordance with the recommendation of the reviewer, the section "Conclusion" has been added to the new version of the article.

The article contains only a small number of abbreviations, so it seems redundant to present a list of them.

Reviewer 2 Report

The present manuscript presents results too preliminary for be accepted for publication in a high standard journal such Molecules.

The major issues include:

- The authors should make the abstract more informative by providing the results of MIC. At least, the MIC for the microorganisms more susceptible to the extract. 
- The authors should highlight in the abstract the most active extract. 

- The manuscript is presented with different text sizes.

- Please provide the ATCC codes for each bacteria in the methods section.

- Please improve the discussion of the results. 

-  The authors could apply Principal component analysis for the evaluation of the chemical constituents of the honey.

- The Antimicrobial effects should be analyzed by other assays, such as time-kill assay, biofilm eradication. 

- The authors should evaluated the cytotoxicity of the extracts.

Author Response

Comment 1: - The authors should make the abstract more informative by providing the results of MIC. At least, the MIC for the microorganisms more susceptible to the extract. 
- The authors should highlight in the abstract the most active extract.

Reply: These remarks were taken into account and the additions recommended by the reviewer were made to the Abstract.

Comment 2: - Please provide the ATCC codes for each bacteria in the methods section

Reply: It's done

Comment 3: The manuscript is presented with different text sizes.

Reply: It's fixed

Comment 3: The authors could apply Principal component analysis for the evaluation of the chemical constituents of the honey.

Reply: This is a misunderstanding. The article does not talk about honey. As for bee venom, there is not enough data for this kind of processing of results.

Comment 4: - The Antimicrobial effects should be analyzed by other assays, such as time-kill assay, biofilm eradication.- The authors should evaluated the cytotoxicity of the extracts.

Reply: We agree that the determination of the action of bee venom aimed at destroying the protective biofilm of bacteria, as well as the cytotoxicity of extracts, could give interesting results. However, this kind of research, requiring a special experimental methodology, was not included in the scope of this study.

Reviewer 3 Report

The paper entitled “Chemical Composition and Antimicrobial Properties of Honey Bee Venom” aimed to determine the composition of volatile and extractive components of dry and fresh bee venom and its antimicrobial activity against several pathogens. Based on the obtained results, authors indicate that the antimicrobial effect of bee venom is associated with the presence of not only peptides but also low molecular weight metabolites.

Page 2 - When talking about the anti-cancer properties of bee venom there is a very nice overview of bee venom in cancer therapy that could be included.

Oršolić N. Bee venom in cancer therapy. Cancer Metastasis Rev. 2012; 31(1-2): 173-94.

Page 2 – When talking about the search for alternative ways to combat cancer one should have in mind the public health impacts of cancer as well such as combinatorial therapy, or using natural products as remedies that could lower the high costs of cancer treatment. Please see:

Viegas S et al. Forgotten public health impacts of cancer - an overview. Arh Hig Rada Toksikol. 2017; 68(4): 287-297.

Authors could discuss the possible toxicity of bee venom to normal non-target cells if used for the treatment. Although there are numerous animal venoms that often show good results in studies there are always open questions regarding venoms’ potential toxicity on normal non-target cells and tissues making this kind of toxicity one of the highest barriers to the possibility on the way to the actual remedy.

Garaj-Vrhovac V, Gajski G. Evaluation of the cytogenetic status of human lymphocytes after exposure to a high concentration of bee venom in vitro. Arh Hig Rada Toksikol. 2009; 60(1): 27-34.

Sjakste N, Gajski G. A Review on Genotoxic and Genoprotective Effects of Biologically Active Compounds of Animal Origin. Toxins (Basel). 2023; 15(2): 165.

Gajski G, Garaj-Vrhovac V. Bee venom induced cytogenetic damage and decreased cell viability in human white blood cells after treatment in vitro: a multi-biomarker approach. Environ Toxicol Pharmacol. 2011; 32(2): 201-11.

Authors could also address the current antibiotic crisis and the scarcity of therapeutic alternatives for the treatment of bacterial infections caused by multi-resistant bacteria where the search for new therapeutic alternatives is persisting challenge with multiple examples of approved or promising AMPs described from various taxa such as melittin and derivatives from bees that are nicely mentioned in a paper published by Bjoern M. von Reumont on modern venomics.

The paper lacks a firmer conclusion section or paragraph. Authors could there address the issues of venom non-specific toxicity which is one of the major obstacles to using them in future treatment along with the above-mentioned challenges of using them as alternatives to known antibiotics. More discussion on the varying content of bee venom in line with the geographical distribution of bees, seasonal variation and/or razing could be mentioned in the conclusion.

Minor remarks:

Abstract – the correct abbreviation would be BV or if HBV is used then I guess honey bee venom should be a full word

Once the abbreviation for BV is introduced please use it consistently throughout the paper

There is different size of the font used throughout the paper that should be unified

Shouldn’t the Materials and methods section be at the end of the paper as per Authors’ guidelines?

Figure 1 – please provide less distorted figures, the font on the chromatogram is too small and distorted and is very hard to read

Author Response

Comment 1: When talking about the anti-cancer properties of bee venom there is a very nice overview of bee venom in cancer therapy that could be included.

Oršolić N. Bee venom in cancer therapy. Cancer Metastasis Rev. 2012; 31(1-2): 173-94.

Page 2 – When talking about the search for alternative ways to combat cancer one should have in mind the public health impacts of cancer as well such as combinatorial therapy, or using natural products as remedies that could lower the high costs of cancer treatment. Please see:

Viegas S et al. Forgotten public health impacts of cancer - an overview. Arh Hig Rada Toksikol. 2017; 68(4): 287-297.

Reply: These suggestions are gratefully accepted. Literary references are included in the article.

Comment 2: Authors could discuss the possible toxicity of bee venom to normal non-target cells if used for the treatment. Although there are numerous animal venoms that often show good results in studies there are always open questions regarding venoms’ potential toxicity on normal non-target cells and tissues making this kind of toxicity one of the highest barriers to the possibility on the way to the actual remedy.

Reply: A discussion of the toxicity of bee venom to healthy cells is beyond the scope of our report. Nevertheless, we mention the importance of this aspect in the "Conclusion" section.

Comment 3: Authors could also address the current antibiotic crisis and the scarcity of therapeutic alternatives for the treatment of bacterial infections caused by multi-resistant bacteria where the search for new therapeutic alternatives is persisting challenge with multiple examples of approved or promising AMPs described from various taxa such as melittin and derivatives from bees that are nicely mentioned in a paper published by Bjoern M. von Reumont on modern venomics.

Reply: Mention of the current "antibiotic crisis" and the desire to find new natural antimicrobial agents that do not cause unwanted side effects is introduced in the article.

If the reviewer is referring to the article by von Reumont et al., published in 2017 in «Toxins», then its content is far from the subject of our publication.

Comment 4: The paper lacks a firmer conclusion section or paragraph. Authors could there address the issues of venom non-specific toxicity which is one of the major obstacles to using them in future treatment along with the above-mentioned challenges of using them as alternatives to known antibiotics. More discussion on the varying content of bee venom in line with the geographical distribution of bees, seasonal variation and/or razing could be mentioned in the conclusion.

Reply: The new version of the article contains such sections and references to aspects of the use of bee venom.

Comment 5: Abstract – the correct abbreviation would be BV or if HBV is used then I guess honey bee venom should be a full word

Once the abbreviation for BV is introduced please use it consistently throughout the paper

There is different size of the font used throughout the paper that should be unified

Reply: All these minor remarks have been taken into account.

Comment 6: Shouldn’t the Materials and methods section be at the end of the paper as per Authors’ guidelines?

Reply: As far as we know, the "Molecules" editors are approaching this issue with flexibility. At least there was no objection to placing the "Materials and Methods" section after the "Introduction" section when our article was published in "Molecules", 2022, 27, 7686.

Comment 7: Figure 1 – please provide less distorted figures, the font on the chromatogram is too small and distorted and is very hard to read.

Reply: There is basically nothing to read in this picture. It serves solely to demonstrate the differences in the chromatogram profile of dry and fresh bee venom.

Round 2

Reviewer 2 Report

The authors have not properly addressed the issues raised for this reviewer. In this case, I do not think it is suitable for publication in Molecules.

Reviewer 3 Report

Authors responded to majority of raised question and improved their paper.